# ADVERSARIALLY REGULARIZED AUTOENCODERS

## ABSTRACT

While autoencoders are a key technique in representation learning for continuous structures, such as images or wave forms, developing general-purpose autoencoders for discrete structures, such as text sequence or discretized images, has proven to be more challenging. In particular, discrete inputs make it more difficult to learn a smooth encoder that preserves the complex local relationships in the input space. In this work, we propose an adversarially regularized autoencoder (ARAE) with the goal of learning more robust discrete-space representations. ARAE jointly trains both a rich discrete-space encoder, such as an RNN, and a simpler continuous space generator function, while using generative adversarial network (GAN) training to constrain the distributions to be similar. This method yields a smoother contracted code space that maps similar inputs to nearby codes, and also an implicit latent variable GAN model for generation. Experiments on text and discretized images demonstrate that the GAN model produces clean interpolations and captures the multimodality of the original space, and that the autoencoder produces improvements in semi-supervised learning as well as state-of-the-art results in unaligned text style transfer task using only a shared continuous-space representation.

## 1 INTRODUCTION

Recent work on regularized autoencoders, such as variational (Kingma & Welling, 2014; Rezende et al., 2014) and denoising (Vincent et al., 2008) variants, has shown significant progress in learning smooth representations of complex, high-dimensional continuous data such as images. These code-space representations facilitate the ability to apply smoother transformations in latent space in order to produce complex modifications of generated outputs, while still remaining on the data manifold.

Unfortunately, learning similar latent representations of discrete structures, such as text sequences or discretized images, remains a challenging problem. Initial work on VAEs for text has shown that optimization is difficult, as the decoder can easily degenerate into a unconditional language model (Bowman et al., 2015b). Recent work on generative adversarial networks (GANs) for text has mostly focused on getting around the use of discrete structures either through policy gradient methods (Che et al., 2017; Hjelm et al., 2017; Yu et al., 2017) or with the Gumbel-Softmax distribution (Kusner & Hernandez-Lobato, 2016). However, neither approach can yet produce robust representations directly.

A major difficulty of discrete autoencoders is mapping a discrete structure to a continuous code vector while also smoothly capturing the complex local relationships of the input space. Inspired by recent work combining pretrained autoencoders with deep latent variable models, we propose to target this issue with an adversarially regularized autoencoder (ARAE). Specifically we jointly train a discrete structure encoder and continuous space generator, while constraining the two models with a discriminator to agree in distribution. This approach allows us to utilize a complex encoder model, such as an RNN, and still constrain it with a very flexible, but more limited generator distribution. The full model can be then used as a smoother discrete structure autoencoder or as a latent variable GAN model where a sample can be decoded, with the same decoder, to a discrete output. Since the system produces a single continuous coded representation—in contrast to methods that act on each RNN state—it can easily be further regularized with problem-specific invariants, for instance to learn to ignore style, sentiment or other attributes for transfer tasks.

Experiments apply ARAE to discretized images and sentences, and demonstrate that the key properties of the model. Using the latent variable model (ARAE-GAN), the model is able to generate varied samples that can be quantitatively shown to cover the input spaces and to generate consistent image and sentence manipulations by moving around in the latent space via interpolation and offset vector

arithmetic. Using the discrete encoder, the model can be used in a semi-supervised setting to give improvement in a sentence inference task. When the ARAE model is trained with task-specific adversarial regularization, the model improves the current best results on sentiment transfer reported in Shen et al. (2017) and produces compelling outputs on a topic transfer task using only a single shared code space. All outputs are listed in the Appendix 9 and code is available at *(removed for review)*.

## 2 RELATED WORK

In practice unregularized autoencoders often learn a degenerate *identity* mapping where the latent code space is free of any structure, so it is necessary to apply some method of regularization. A popular approach is to regularize through an explicit prior on the code space and use a variational approximation to the posterior, leading to a family of models called variational autoencoders (VAE) (Kingma & Welling, 2014; Rezende et al., 2014). Unfortunately VAEs for discrete text sequences can be challenging to train—for example, if the training procedure is not carefully tuned with techniques like word dropout and KL annealing (Bowman et al., 2015b), the decoder simply becomes a language model and ignores the latent code (although there has been some recent successes with convolutional models (Semeniuta et al., 2017; Yang et al., 2017)). One possible reason for the difficulty in training VAEs is due to the strictness of the prior (usually a spherical Gaussian) and/or the parameterization of the posterior. There has been some work on making the prior/posterior more flexible through explicit parameterization (Rezende & Mohamed, 2015; Kingma et al., 2016; Chen et al., 2017). A notable technique is adversarial autoencoders (AAE) (Makhzani et al., 2015) which attempt to imbue the model with a more flexible prior implicitly through adversarial training. In AAE framework, the discriminator is trained to distinguish between samples from a fixed prior distribution and the input encoding, thereby pushing the code distribution to match the prior. While this adds more flexibility, it has similar issues for modeling text sequences and suffers from mode-collapse in our experiments. Our approach has similar motivation, but notably we do not sample from a fixed prior distribution—our 'prior' is instead parameterized through a flexible generator. Nonetheless, this view (which has been observed by various researchers (Tran et al., 2017; Mescheder et al., 2017; Makhzani & Frey, 2017)) provides an interesting connection between VAEs and GANs.

The success of GANs on images have led many researchers to consider applying GANs to discrete data such as text. Policy gradient methods are a natural way to deal with the resulting non-differentiable generator objective when training directly in discrete space (Glynn, 1987; Williams, 1992). When trained on text data however, such methods often require pre-training/co-training with a maximum likelihood (i.e. language modeling) objective (Che et al., 2017; Yu et al., 2017; Li et al., 2017). This precludes there being a latent encoding of the sentence, and is also a potential disadvantage of existing language models (which can otherwise generate locally-coherent samples). Another direction of work has been through reparameterizing the categorical distribution with the Gumbel-Softmax trick (Jang et al., 2017; Maddison et al., 2017)—while initial experiments were encouraging on a synthetic task (Kusner & Hernandez-Lobato, 2016), scaling them to work on natural language is a challenging open problem. There has also been a flurry of recent, related approaches that work directly with the soft outputs from a generator (Gulrajani et al., 2017; Sai Rajeswar, 2017; Shen et al., 2017; Press et al., 2017). For example, Shen et al. (Shen et al., 2017) exploits adversarial loss for unaligned style transfer between text by having the discriminator act on the RNN hidden states and using the soft outputs at each step as input to an RNN generator, utilizing the Professor-forcing framework (Lamb et al., 2016). Our approach instead works entirely in code space and does not require utilizing RNN hidden states directly.

## 3 BACKGROUND

**Discrete Structure Autoencoders** Define $\mathcal{X} = \mathcal{V}^n$ to be a set of discrete structures where $\mathcal{V}$ is a vocabulary of symbols and $\mathbb{P}_x$ to be a distribution over this space. For instance, for binarized images $\mathcal{V} = \{0, 1\}$ and $n$ is the number of pixels, while for sentences $\mathcal{V}$ is the vocabulary and $n$ is the sentence length. A discrete autoencoder consists of two parameterized functions: a deterministic encoder function $\text{enc}_\phi : \mathcal{X} \mapsto \mathcal{C}$ with parameters $\phi$ that maps from input to code space and a conditional decoder distribution $p_\psi(\mathbf{x} \mid \mathbf{c})$ over structures $\mathcal{X}$ with parameters $\psi$. The parameters are trained on a cross-entropy reconstruction loss:

$$\mathcal{L}_{\text{rec}}(\phi, \psi) = -\log p_\psi(\mathbf{x} \mid \text{enc}_\phi(\mathbf{x}))$$

The choice of the encoder and decoder parameterization is specific to the structure of interest, for example we use RNNs for sequences. We use the notation, $\hat{\mathbf{x}} = \arg\max_{\mathbf{x}} p_\psi(\mathbf{x} \mid \text{enc}_\phi(\mathbf{x}))$ for the (approximate) decoder mode. When $\mathbf{x} = \hat{\mathbf{x}}$ the autoencoder is said to perfectly reconstruct $\mathbf{x}$.

**Generative Adversarial Networks** GANs are a class of parameterized implicit generative models (Goodfellow et al., 2014). The method approximates drawing samples from a true distribution $\mathbf{c} \sim \mathbb{P}_r$ by instead employing a latent variable $\mathbf{z}$ and a parameterized deterministic generator function $\tilde{\mathbf{c}} = g_\theta(\mathbf{z})$ to produce samples $\tilde{\mathbf{c}} \sim \mathbb{P}_g$. Initial work on GANs minimizes the Jensen-Shannon divergence between the distributions. Recent work on Wasserstein GAN (WGAN) (Arjovsky et al., 2017), replaces this with the *Earth-Mover* (Wasserstein-1) distance.

GAN training utilizes two separate models: a *generator* $g_\theta(\mathbf{z})$ maps a latent vector from some easy-to-sample source distribution to a sample and a critic/discriminator $f_w(\mathbf{c})$ aims to distinguish *real* data and *generated* samples from $g_\theta$. Informally, the generator is trained to fool the critic, and the critic to tell real from generated. WGAN training uses the following min-max optimization over generator parameters $\theta$ and critic parameters $w$,

$$\min_\theta \max_{w \in \mathcal{W}} \mathbb{E}_{\mathbf{c} \sim \mathbb{P}_r}[f_w(\mathbf{c})] - \mathbb{E}_{\tilde{\mathbf{c}} \sim \mathbb{P}_g}[f_w(\tilde{\mathbf{c}})], \tag{1}$$

where $f_w : \mathcal{C} \mapsto \mathbb{R}$ denotes the critic function, $\tilde{\mathbf{c}}$ is obtained from the generator, $\tilde{\mathbf{c}} = g_\theta(\mathbf{z})$, and $\mathbb{P}_r$ and $\mathbb{P}_g$ are real and generated distributions. If the critic parameters $w$ are restricted to an 1-Lipschitz function set $\mathcal{W}$, this term correspond to minimizing Wasserstein-1 distance $W(\mathbb{P}_r, \mathbb{P}_g)$. We use a naive approximation to enforce this property by weight-clipping, i.e. $w = [-\epsilon, \epsilon]^d$ (Arjovsky et al., 2017).

## 4 MODEL: ADVERSARIALLY REGULARIZED AUTOENCODER

Ideally, a discrete autoencoder should be able to *reconstruct* $x$ from $c$, but also *smoothly* assign similar codes $c$ and $c'$ to similar $x$ and $x'$. For continuous autoencoders, this property can be enforced directly through explicit regularization. For instance, contractive autoencoders (Rifai et al., 2011) regularize their loss by the functional smoothness of $\text{enc}_\phi$. However, this criteria does not apply when inputs are discrete and we lack even a metric on the input space. How can we enforce that similar discrete structures map to nearby codes?

Adversarially regularized autoencoders target this issue by learning a parallel continuous-space generator with a restricted functional form to act as a smoother reference encoding. The joint objective regularizes the autoencoder to constrain the discrete encoder to agree in distribution with its continuous counterpart:

$$\min_{\phi, \psi, \theta} \quad \mathcal{L}_{\text{rec}}(\phi, \psi) + \lambda^{(1)} W(\mathbb{P}_r, \mathbb{P}_g)$$

Above $W$ is the Wasserstein-1 distance between $\mathbb{P}_r$ the distribution of codes from the discrete encoder model ($\text{enc}_\phi(x)$ where $x \sim \mathbb{P}(x)$) and $\mathbb{P}_g$ is the distribution of codes from the continuous generator model ($g_\theta(z)$ for some $z$, e.g. $z \sim \mathcal{N}(0, I)$). To approximate Wasserstein-1 term, the $W$ function includes an embedded critic function which is optimized adversarially to the encoder and generator as described in the background. The full model is shown in Figure 1.

To train the model, we use a block coordinate descent to alternate between optimizing different parts of the model: (1) the encoder and decoder to minimize reconstruction loss, (2) the WGAN critic function to approximate the $W$ term, (3) the encoder and generator to adversarially fool the critic to minimize $W$:

$$1) \min_{\phi, \psi} \qquad \mathcal{L}_{\text{rec}}(\phi, \psi)$$

$$2) \min_{w \in \mathcal{W}} \qquad \mathcal{L}_{\text{cri}}(w) = \qquad \max_{w \in \mathcal{W}} \quad \mathbb{E}_{\mathbf{x} \sim \mathbb{P}_x}[f_w(\text{enc}_\phi(\mathbf{x}))] - \mathbb{E}_{\tilde{\mathbf{c}} \sim \mathbb{P}_g}[f_w(\tilde{\mathbf{c}})]$$

$$3) \min_{\phi, \theta} \qquad \mathcal{L}_{\text{encs}}(\phi, \theta) = \qquad \min_{\phi, \theta} \quad \mathbb{E}_{\mathbf{x} \sim \mathbb{P}_x}[f_w(\text{enc}_\phi(\mathbf{x}))] - \mathbb{E}_{\tilde{\mathbf{c}} \sim \mathbb{P}_g}[f_w(\tilde{\mathbf{c}})]$$

The full training algorithm is shown in Algorithm 1.

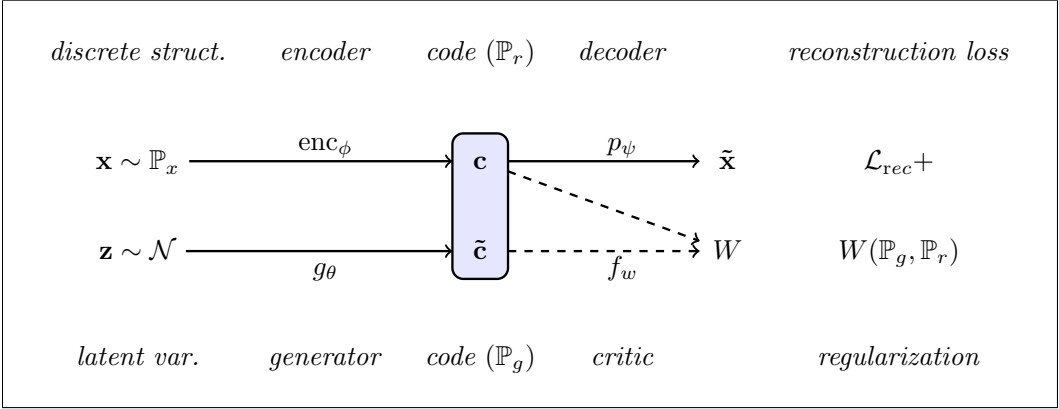

Figure 1: ARAE architecture. The model can be used as an autoencoder, where a structure $\mathbf{x}$ is encoded and decoded to produce $\hat{\mathbf{x}}$, and as a GAN (ARAE-GAN), where a sample $\mathbf{z}$ is passed though a generator $g_\theta$ to produce a code vector, which is similarly decoded to $\tilde{\mathbf{x}}$. The critic function $f_w$ is only used at training to help approximate $W$.

---

**Algorithm 1** ARAE Training

---

**for** number of training iterations **do**
    *(1) Train the autoencoder for reconstruction* [$\mathcal{L}_{rec}(\phi, \psi)$].
        Sample $\{\mathbf{x}^{(i)}\}_{i=1}^m \sim \mathbb{P}_x$ and compute code-vectors $\mathbf{c}^{(i)} = enc_\phi(\mathbf{x}^{(i)})$.
        Backpropagate reconstruction loss, $\mathcal{L}_{rec} = -\frac{1}{m}\sum_{i=1}^m \log p_\psi(\mathbf{x}^{(i)} \mid \mathbf{c}^{(i)}, [\mathbf{y}^{(i)}])$, and update.
    *(2) Train the critic* [$\mathcal{L}_{cri}(w)$] (Repeat k times)
        Sample $\{\mathbf{x}^{(i)}\}_{i=1}^m \sim \mathbb{P}_x$ and $\{\mathbf{z}^{(i)}\}_{i=1}^m \sim \mathcal{N}(0, \mathbf{I})$.
        Compute code-vectors $\mathbf{c}^{(i)} = enc_\phi(\mathbf{x}^{(i)})$ and $\tilde{\mathbf{c}}^{(i)} = g_\theta(\mathbf{z}^{(i)})$.
        Backpropagate loss $-\frac{1}{m}\sum_{i=1}^m f_w(\mathbf{c}^{(i)}) + \frac{1}{m}\sum_{i=1}^m f_w(\tilde{\mathbf{c}}^{(i)})$, update, clip the critic $w$ to $[-\epsilon, \epsilon]^d$.
    *(3) Train the generator and encoder adversarially to critic* [$\mathcal{L}_{encs}(\phi, \theta)$]
        Sample $\{\mathbf{x}^{(i)}\}_{i=1}^m \sim \mathbb{P}_x$ and $\{\mathbf{z}^{(i)}\}_{i=1}^m \sim \mathcal{N}(0, \mathbf{I})$
        Compute code-vectors $\mathbf{c}^{(i)} = enc_\phi(\mathbf{x}^{(i)})$ and $\tilde{\mathbf{c}}^{(i)} = g_\theta(\mathbf{z}^{(i)})$.
        Backpropagate adversarial loss $\frac{1}{m}\sum_{i=1}^m f_w(\mathbf{c}^{(i)}) - \frac{1}{m}\sum_{i=1}^m f_w(\tilde{\mathbf{c}}^{(i)})$ and update.

---

**Extension: Code Space Transfer** One benefit of the ARAE framework is that it compresses the input to a single code vector. This framework makes it ideal for manipulating discrete objects while in continuous code space. For example, consider the problem of unaligned transfer, where we want to change an attribute of a discrete input without supervised examples, e.g. to change the topic or sentiment of a sentence. First, we extend the decoder to condition on a transfer variable denoting this attribute $\mathbf{y}$ which is known during training, to learn $p_\psi(\mathbf{x} \mid \mathbf{c}, y)$. Next, we train the code space to be invariant to this attribute, to force it to be learned fully by the decoder. Specifically, we further regularize the code space to map similar $x$ with different attribute labels $y$ near enough to fool a code space attribute classifier, i.e.:

$$\min_{\phi, \psi, \theta} \quad \mathcal{L}_{rec}(\phi, \psi) + \lambda^{(1)} W(\mathbb{P}_r, \mathbb{P}_g) - \lambda^{(2)} \mathcal{L}_{class}(\phi, u)$$

where $\mathcal{L}_{class}(\phi, u)$ is the loss of a classifier $p_u(y \mid \mathbf{c})$ from code space to labels (in our experiments we always set $\lambda^{(2)} = 1$). To incorporate this additional regularization, we simply add two more gradient update steps: (2b) training a classifier to discriminate codes, and (3b) adversarially training the encoder to fool this classifier. The algorithm is shown in Algorithm 2. Note that similar technique has been introduced in other domains, notably in images (Lample et al., 2017) and video modeling (Denton & Birodkar, 2017).

## 5 METHODS AND ARCHITECTURES

We experiment with three different ARAE models: (1) an autoencoder for discretized images trained on the binarized version of MNIST, (2) an autoencoder for text sequences trained using the Stanford Natural Language Inference (SNLI) corpus (Bowman et al., 2015a), and (3) an autoencoder trained

---

**Algorithm 2** ARAE Transfer Extension

---

[Each loop additionally:]

**(2b) Train the code classifier** [$\min_u \mathcal{L}_{\text{class}}(\phi, u)$]

      Sample $\{\mathbf{x}^{(i)}\}_{i=1}^{m} \sim \mathbb{P}_x$, lookup $y^{(i)}$, and compute code-vectors $\mathbf{c}^{(i)} = \text{enc}_\phi(\mathbf{x}^{(i)})$.

      Backpropagate loss $-\frac{1}{m}\sum_{i=1}^{m} \log p_u(y^{(i)}|\mathbf{c}^{(i)})$, update.

**(3b) Train the encoder adversarially to code classifier** [$\max_\phi \mathcal{L}_{\text{class}}(\phi, u)$]

      Sample $\{\mathbf{x}^{(i)}\}_{i=1}^{m} \sim \mathbb{P}_x$, lookup $y^{(i)}$, and compute code-vectors $\mathbf{c}^{(i)} = \text{enc}_\phi(\mathbf{x}^{(i)})$.

      Backpropagate adversarial classifier loss $-\frac{1}{m}\sum_{i=1}^{m} \log p_u(1 - y^{(i)} \mid \mathbf{c}^{(i)})$, update.

---

for text transfer (Section 6.2) based on the Yelp and Yahoo datasets for unaligned sentiment and topic transfer. All three models utilize the same generator architecture, $g_\theta$. The generator architecture uses a low dimensional $\mathbf{z}$ with a Gaussian prior $p(\mathbf{z}) = \mathcal{N}(0, \mathbf{I})$, and maps it to $\mathbf{c}$. Both the critic $f_w$ and the generator $g_\theta$ are parameterized as feed-forward MLPs.

The *image* model uses fully-connected NN to autoencode binarized images. Here $\mathcal{X} = \{0, 1\}^n$ where $n$ is the image size. The encoder used is a feed-forward MLP network mapping from $\{0, 1\}^n \mapsto \mathbb{R}^m$, $\text{enc}_\phi(\mathbf{x}) = \text{MLP}(\mathbf{x}; \phi) = \mathbf{c}$. The decoder predicts each pixel in $\mathbf{x}$ as a parameterized logistic regression, $p_\psi(\mathbf{x} \mid \mathbf{c}) = \prod_{j=1}^{n} \sigma(\mathbf{h})^{x_j}(1 - \sigma(\mathbf{h}))^{1-x_j}$ where $\mathbf{h} = \text{MLP}(\mathbf{c}; \psi)$.

The *text* model uses a recurrent neural network (RNN) for both the encoder and decoder. Here $\mathcal{X} = \mathcal{V}^n$ where $n$ is the sentence length and $\mathcal{V}$ is the vocabulary of the underlying language. Define an RNN as a parameterized recurrent function $\mathbf{h}_j = \text{RNN}(x_j, \mathbf{h}_{j-1}; \phi)$ for $j = 1 \ldots n$ (with $\mathbf{h}_0 = \mathbf{0}$) that maps a discrete input structure $\mathbf{x}$ to hidden vectors $\mathbf{h}_1 \ldots \mathbf{h}_n$. For the encoder, we define $\text{enc}_\phi(\mathbf{x}) = \mathbf{h}_n = \mathbf{c}$. For decoding we feed $\mathbf{c}$ as an additional input to the decoder RNN at each time step, i.e. $\tilde{\mathbf{h}}_j = \text{RNN}(x_j, \tilde{\mathbf{h}}_{j-1}, \mathbf{c}; \psi)$, and further calculate the distribution over $\mathcal{V}$ at each time step via softmax, $p_\psi(\mathbf{x} \mid \mathbf{c}) = \prod_{j=1}^{n} \text{softmax}(\mathbf{W}\tilde{\mathbf{h}}_j + \mathbf{b})_{x_j}$ where $\mathbf{W}$ and $\mathbf{b}$ are parameters (part of $\psi$). Finding the most likely sequence $\tilde{\mathbf{x}}$ under this distribution is intractable, but it is possible to approximate it using greedy search or beam search. In our experiments we use an LSTM architecture (Hochreiter & Schmidhuber, 1997) for both the encoder/decoder and decode using greedy search. The *text transfer* model uses the same architecture as the text model but extends it with a code space classifier $p(y|\mathbf{c})$ which is modeled using an MLP and trained to minimize cross-entropy.

Our baselines utilize a standard autoencoder (AE) and the cross-aligned autoencoder (Shen et al., 2017) for transfer. Note that in both our ARAE and standard AE experiments, the encoded code from the encoder is normalized to lie on the unit sphere, and the generated code is bounded to lie in $(-1, 1)^n$ by the $\tanh$ function at output layer. We additionally experimented with the sequence VAE introduced by Bowman et al. (2015b) and the adversarial autoencoder (AAE) model (Makhzani et al., 2015) on the SNLI dataset. However despite extensive parameter tuning we found that neither model was able to learn meaningful latent representations—the VAE simply ignored the latent code and the AAE experienced mode-collapse and repeatedly generated the same samples. The Appendix 12 includes detailed descriptions of the hyperparameters, model architecture, and training regimes.

## 6 EXPERIMENTS

Our experiments consider three aspects of the model. First we measure the empirical impact of regularization on the autoencoder. Next we apply the discrete autoencoder to two applications, unaligned style transfer and semi-supervised learning. Finally we employ the learned generator network as an implicit latent variable model (ARAE-GAN) over discrete sequences.

### 6.1 IMPACT OF REGULARIZATION ON DISCRETE ENCODING

Our main goal for ARAE is to regularize the model produce a smoother encoder by requiring the distribution from the encoder to match the distribution from the continuous generator over a simple latent variable. To examine this claim we consider two basic statistical properties of the code space during training of the text model on SNLI, shown in Figure 2. On the left, we see that the $\ell 2$ norm of $\mathbf{c}$ and code $\tilde{\mathbf{c}}$ converge quickly in ARAE training. The encoder code is always restricted to be on the unit sphere, and the generated code $\tilde{\mathbf{c}}$ quickly learns to match it. The middle plot shows the convergence of the trace of the covariance matrix between the generator and the encoder as training

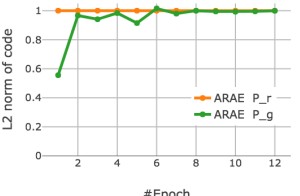 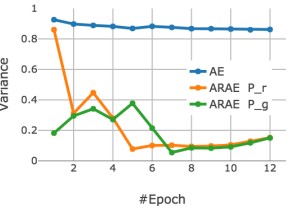 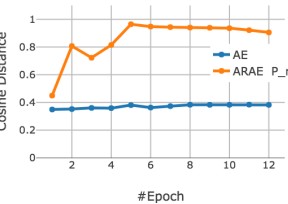

Figure 2: Left: $\ell 2$ norm of encoder code $\mathbf{c}$ and generator code $\tilde{\mathbf{c}}$ during ARAE training. The encoder $\mathbf{c}$ is normalized by the model, whereas the generator learns to match this as training progresses. Middle: Sum of the dimension-wise variances of the encoder codes $\mathbb{P}_r$ and generator codes $\mathbb{P}_g$ compared to that of the standard AE. Right: Average cosine similarity of nearby sentences (edit-distance wise) for the ARAE and AE.

| $k$ | AE | ARAE | | | | | |
|---|---|---|---|---|---|---|---|
| | | | Original | A woman wearing sunglasses . | | Original | They have been swimming . |
| | | | Noised | A woman sunglasses wearing . | | Noised | been have They swimming . |
| 0 | 1.06 | 2.19 | AE | A woman sunglasses wearing sunglasses . | | AE | been have been swimming . |
| 1 | 4.51 | 4.07 | ARAE | A woman wearing sunglasses . | | ARAE | Children have been swimming . |
| 2 | 6.61 | 5.39 | Original | Pets galloping down the street . | | Original | The child is sleeping . |
| 3 | 9.14 | 6.86 | Noised | Pets down the galloping street . | | Noised | child The is sleeping . |
| | | | AE | Pets riding the down galloping . | | AE | The child is sleeping is . |
| 4 | 9.97 | 7.47 | ARAE | Pets congregate down the street near a ravine . | | ARAE | The child is sleeping . |

Table 1: **Left**. Reconstruction error (negative log-likelihood averaged over sentences) of the original sentence from a corrupted sentence. Here $k$ is the number of swaps performed on the original sentence. **Right**. Samples generated from AE and ARAE where the input is noised by swapping words.

progresses. We find that variance of the encoder and the generator match after several epochs. To check the smoothness of the model, for both ARAE/AE, we take a sentence and calculate the average cosine similarity of 100 randomly-selected sentences that had an edit-distance of at most 5 to the original sentence. We do this for 250 sentences and calculate the mean of the average cosine similarity. Figure 2 (right) shows that the cosine similarity of nearby sentences is quite high for the ARAE than in the case for the AE. Edit-distance is not an ideal proxy for similarity in sentences, but it is often a sufficient condition.

Finally an ideal representation should be robust to small changes of the input around the training examples in code space (Rifai et al., 2011). We can test this property by feeding a noised input to the encoder and (i) calculating the score given to the original input, and (ii) checking the reconstructions. Table 1 (right) shows an experiment for text where we add noise by permuting $k$ words in each sentence. We observe that the ARAE is able to map a noised sentence to a natural sentence, (though not necessarily the denoised sentence). Table 1 (left) shows empirical results for these experiments. We obtain the reconstruction error (i.e. negative log likelihood) of the original (non-noised) sentence under the decoder, utilizing the noised code. We find that when $k = 0$ (i.e. no swaps), the regular AE better reconstructs the input as expected. However, as we increase the number of swaps and push the input further away from the data manifold, the ARAE is more likely to produce the original sentence. We note that unlike denoising autoencoders which require a domain-specific noising function (Hill et al., 2016; Vincent et al., 2008), the ARAE is not explicitly trained to denoise an input, but learns to do so as a byproduct of adversarial regularization.

## 6.2 APPLICATIONS OF DISCRETE AUTOENCODER

**Unaligned Text Transfer** A smooth autoencoder combined with low reconstruction error should make it possible to more robustly manipulate discrete objects through code space without dropping off the data manifold. To test this hypothesis, we experimented with two unaligned text transfer tasks. For these tasks, we attempt to change one attribute of a sentence without aligned examples of this change. To perform this transfer, we learn a code space that can represent an input that is agnostic to this attribute, and a decoder that can incorporate the attribute (as described in Section 4). We experiment with unaligned transfer of sentiment on the Yelp corpus and topic on the Yahoo corpus (Zhang et al., 2015).

| Model | Automatic Evaluation | | | | Human Evaluation | | |
|---|---|---|---|---|---|---|---|
| | Transfer | BLEU | PPL | Reverse PPL | Transfer | Similarity | Naturalness |
| Cross-Aligned AE | 77.1% | 17.75 | 65.9 | 124.2 | 57% | 3.8 | 2.7 |
| AE | 59.3% | 37.28 | 31.9 | 68.9 | - | - | - |
| ARAE, $\lambda_a^{(1)}$ | 73.4% | 31.15 | 29.7 | 70.1 | - | - | - |
| ARAE, $\lambda_b^{(1)}$ | 81.8% | 20.18 | 27.7 | 77.0 | 74% | 3.7 | 3.8 |

Table 2: Experiments on sentiment transfer. Left shows the automatic metrics (Transfer/BLEU/PPL/Reverse PPL) while right shows human evaluation metrics (Transfer/Similarity/Naturalness). Cross-Aligned AE is from Shen et al. (2017)

| | Positive $\Rightarrow$ Negative | | Negative $\Rightarrow$ Positive |
|---|---|---|---|
| | great indoor mall . | | hell no ! |
| ARAE | no smoking mall . | ARAE | hell great ! |
| Cross-AE | terrible outdoor urine . | Cross-AE | incredible pork ! |
| | it has a great atmosphere , with wonderful service . | | small , smokey , dark and rude management . |
| ARAE | it has no taste , with a complete jerk . | ARAE | small , intimate , and cozy friendly staff . |
| Cross-AE | it has a great horrible food and run out service . | Cross-AE | great , , , chips and wine . |
| | we came on the recommendation of a bell boy and the food was amazing . | | the people who ordered off the menu did n't seem to do much better . |
| ARAE | we came on the recommendation and the food was a joke . | ARAE | the people who work there are super friendly and the menu is good . |
| Cross-AE | we went on the car of the time and the chicken was awful . | Cross-AE | the place , one of the office is always worth you do a business . |

Table 3: Sentiment transfer results. Original sentence and transferred output (from ARAE and the Cross-Aligned AE) of 6 randomly-drawn examples.

For sentiment we follow the same setup as Shen et al. (2017) and split the Yelp corpus into two sets of unaligned positive and negative reviews. We train an ARAE as an autoencoder with two separate decoders, one for positive and one for negative sentiment, and incorporate adversarial training of the encoder to remove sentiment information from the code space. We test by encoding in sentences of one class and decoding, greedily, with the opposite decoder.

Our evaluation is based on four automatic metrics, shown in Table 2: (i) Transfer: measuring how successful the model is at transferring sentiment based on an automatic classifier (we use the `fastText` library (Joulin et al., 2016)). (ii) BLEU: measuring the consistency between the transferred text and the original. We expect the model to maintain as much information as possible and transfer only the style; (iii) Perplexity: measuring the fluency of the generated text; (iv) Reverse Perplexity: measuring the extent to which the generations are representative of the underlying data distribution.[1] Both perplexity numbers are obtained by training an RNN language model.

We additionally perform human evaluations on the cross-aligned AE and our best ARAE model. We randomly select 1000 sentences (500/500 positive/negative), obtain the corresponding transfers from both models, and ask Amazon Mechanical Turkers to evaluate the sentiment (Positive/Neutral/Negative) and naturalness (1-5, 5 being most natural) of the transferred sentences. We create a separate task in which we show the Turkers the original and the transferred sentences, and ask them to evaluate the similarity based on sentence structure (1-5, 5 being most similar). We explicitly ask the Turkers to disregard sentiment in their similarity assessment.

In addition to comparing against the cross-aligned AE of Shen et al. (2017), we also compare against a vanilla AE trained without adversarial regularization. For ARAE, we experimented with different $\lambda^{(1)}$ weighting on the adversarial loss (see section 4) with $\lambda_a^{(1)} = 1, \lambda_b^{(1)} = 10$. We generally set $\lambda^{(2)} = 1$. Experimentally the adversarial regularization enhances transfer and perplexity, but tends to make the transferred text less similar to the original, compared to the AE. Some randomly selected sentences are shown in figure 6 and more samples are shown available in Appendix 9.

The same method can be applied to other style transfer tasks, for instance the more challenging Yahoo QA data (Zhang et al., 2015). For Yahoo we chose 3 relatively distinct topic classes for transfer: Science & Math, Entertainment & Music, and Politics & Government. As the dataset contains both

---

[1]This *reverse perplexity* is calculated by training a language model on the generated data and measuring perplexity on held-out, real data (i.e. reverse of regular perplexity). We also found this metric to be helpful for early-stopping based on validation data.

questions and answers, we separated our experiments into titles (questions) and replies (answers). The qualitative results are showed in table 4. See Appendix 9 for additional generation examples.

| | Original Science | | Original Music | | Original Politics |
|---|---|---|---|---|---|
| | what is an event horizon with regards to black holes ? | | do you know a website that you can find people who want to join bands ? | | republicans : would you vote for a cheney / satan ticket in 2008 ? |
| Music | what is your favorite sitcom with adam sandler ? | Science | do you know a website that can help me with science ? | Science | guys : how would you solve this question ? |
| Politics | what is an event with black people ? | Politics | do you think that you can find a person who is in prison ? | Music | guys : would you rather be a good movie ? |
| | take 1ml of hcl ( concentrated ) and dilute it to 50ml . | | all three are fabulous artists , with just incredible talent ! ! | | 4 years of an idiot in office + electing the idiot again = ! |
| Music | take em to you and shout it to me | Science | all three are genetically bonded with water , but just as many substances , are capable of producing a special case . | Science | 4 years of an idiot in the office of science ? |
| Politics | take bribes to islam and it will be punished . | Politics | all three are competing with the government , just as far as i can . | Music | 4 ) <unk> in an idiot , the idiot is the best of the two points ever ! |
| | just multiply the numerator of one fraction by that of the other . | | but there are so many more i can 't think of ! | | anyone who doesnt have a billion dollars for all the publicity cant win . |
| Music | just multiply the fraction of the other one that 's just like it . | Science | but there are so many more of the number of questions . | Science | anyone who doesnt have a decent chance is the same for all the other . |
| Politics | just multiply the same fraction of other countries . | Politics | but there are so many more of the can i think of today . | Music | anyone who doesnt have a lot of the show for the publicity . |

Table 4: Random samples from Yahoo topic transfer. Note the first row is from ARAE trained on titles while the following ones are from replies.

**Semi-Supervised Training** We further utilize ARAE in a standard AE setup for semi-supervised training. We experiment on a natural language inference task, shown in Table 5 (right). We use 22.2%, 10.8% and 5.25% of the original labeled training data, and use the rest of the training set for unlabeled training. The labeled set is randomly picked. The full SNLI training set contains 543k sentence pairs, and we use supervised sets of 120k, 59k and 28k sentence pairs respectively for the three settings. As a baseline we use an AE trained on the additional data, similar to the setting explored in Dai & Le (2015). For ARAE we use the subset of unsupervised data of length $< 15$, which roughly includes 655k single sentences (due to the length restriction, this is a subset of 715k sentences that were used for AE training). As observed by Dai & Le (2015), training on unlabeled data with an AE objective improves upon a model just trained on labeled data. Training with adversarial regularization provides further gains.

### 6.3 A Latent Variable Model for Discrete Structures

After training, an ARAE can also be used as an implicit latent variable model controlled by $\mathbf{z}$ and the generator $g_\theta$, which we refer to as ARAE-GAN. While models of this form have been widely used for generation in other modalities, they have been less effective for discrete structures. In this section, we attempt to measure the effectiveness of this induced discrete GAN.

A common test for a GANs ability mimic the true distribution $\mathbb{P}_r$ is to train a simple model on generated samples from $\mathbb{P}_g$. While there are pitfalls of this evaluation (Theis et al., 2016), it provides a starting point for text modeling. Here we generate 100k samples from (i) ARAE-GAN, (ii) an AE[2], (iii) a RNN LM trained on the same data, and (iv) the real training set (samples from the models are

---

[2]To "sample" from an AE we fit a multivariate Gaussian to the code space after training and generate code vectors from this Gaussian to decode back into sentence space.

| Model | Medium | Small | Tiny |
|---|---|---|---|
| Supervised Encoder | 65.9% | 62.5% | 57.9% |
| Semi-Supervised AE | 68.5% | 64.6% | 59.9% |
| Semi-Supervised ARAE | 70.9% | 66.8% | 62.5% |

| Data for LM | Reverse PPL |
|---|---|
| Real data | 27.4 |
| LM samples | 90.6 |
| AE samples | 97.3 |
| ARAE-GAN samples | 82.2 |

Table 5: **Left.** Semi-Supervised accuracy on the natural language inference (SNLI) test set, respectively using 22.2% (medium), 10.8% (small), 5.25% (tiny) of the supervised labels of the full SNLI training set (rest used for unlabeled AE training). **Right.** Perplexity (lower is better) of language models trained on the synthetic samples from a GAN/AE/LM, and evaluated on real data (Reverse PPL).

A man is on the corner in a sport area .
A man is on corner in a road all .
A lady is on outside a racetrack .
A lady is outside on a racetrack .
A lot of people is outdoors in an urban setting .
A lot of people is outdoors in an urban setting .
A lot of people is outdoors in an urban setting .

A man is on a ship path with the woman .
A man is on a ship path with the woman .
A man is passing on a bridge with the girl .
A man is passing on a bridge with the girl .
A man is passing on a bridge with the girl .
A man is passing on a bridge with the dogs .
A man is passing on a bridge with the dogs .

A man in a cave is used an escalator .
A man in a cave is used an escalator
A man in a cave is used chairs .
A man in a number is used many equipment
A man in a number is posing so on a big rock .
People are posing in a rural area .
People are posing in a rural area.

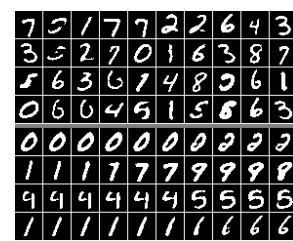

Figure 3: Sample interpolations from the ARAE-GAN. Constructed by linearly interpolating in the latent space and decoding to the output space. Word changes are highlighted in black. Results of the ARAE. The top block shows output generation of the decoder taking *fake* hidden codes generated by the GAN; the bottom block shows sample interpolation results.

| Transform | Match % | Prec |
|---|---|---|
| walking | 85 | 79.5 |
| man | 92 | 80.2 |
| two | 86 | 74.1 |
| dog | 88 | 77.0 |
| standing | 89 | 79.3 |
| several | 70 | 67.0 |

A man in a tie is sleeping and clapping on balloons .
A person is standing in the air beneath a criminal .
The jewish boy is trying to stay out of his skateboard .
The people works in a new uniform studio .
Some child head a playing plastic with drink .
A baby workers is watching steak with the water .
The people shine or looks into an area .
The boy 's babies is wearing a huge factory .
A women are walking outside near a man .
The dogs are sleeping in front of the dinner .
A side child listening to a piece with steps playing on a table .
Two children are working in red shirt at the cold field .

$\Rightarrow$walking A man in a tie is clapping and walking dogs .
$\Rightarrow$walking A person is walking in the air beneath a pickup .
$\Rightarrow$man The jewish man is trying to stay out of his horse .
$\Rightarrow$man A man works in a new studio uniform .
$\Rightarrow$Two Two children playing a head with plastic drink .
$\Rightarrow$Two Two workers watching baby steak with the grass .
$\Rightarrow$dog The dog arrives or looks into an area .
$\Rightarrow$dog The dog 's babies is wearing a huge ears .
$\Rightarrow$standing Three women are standing near a man walking .
$\Rightarrow$standing Two dogs are standing in front of the dinner .
$\Rightarrow$Several Several child playing a guitar on side with a table .
$\Rightarrow$Several Several children working in red shirt are cold at the field .

Figure 4: **Left**. Quantitative evaluation of transformations. Match % refers to the % of samples where at least one decoder samples (per 100) had the desired transformation in the output, while Prec. measures the average precision of the output against the original sentence. **Right**. Examples (out of 100 decoder samples per sentence) where the offset vectors produced successful transformations of the original sentence. See Appendix 11 for full methodology.

shown in Appendix 10). All models are of the same size to allow for fair comparison. We train an RNN language model on generated samples and evaluate on held-out data to calculate the *reverse perplexity*. As can be seen from Table 5, training on real data (understandably) outperforms training on generated data by a large margin. Surprisingly however, we find that a language model trained on ARAE-GAN data performs slightly better than one trained on LM-generated/AE-generated data. We further found that the reverse PPL of an AAE (Makhzani et al., 2015) was quite high (980) due to mode-collapse.

Another property of GANs (and VAEs) is that the Gaussian form of $\mathbf{z}$ induces the ability to smoothly interpolate between outputs by exploiting the structure of the latent space. While language models may provide a better estimate of the underlying probability space, constructing this style of interpolation would require combinatorial search, which makes this a useful feature of text GANs. We experiment with this property by sampling two points $\mathbf{z}_0$ and $\mathbf{z}_1$ from $p(\mathbf{z})$ and constructing intermediary points $\mathbf{z}_\lambda = \lambda\mathbf{z}_1 + (1-\lambda)\mathbf{z}_0$. For each we generate the argmax output $\tilde{\mathbf{x}}_\lambda$. The samples are shown in Figure 3 (left) for text and in Figure 3 (right) for a discretized MNIST ARAE-GAN.

A final intriguing property of image GANs is the ability to move in the latent space via offset vectors (similar to the case with word vectors (Mikolov et al., 2013)). For example, Radford et al. (Radford et al., 2016) observe that when the mean latent vector for "men with glasses" is subtracted from the mean latent vector for "men without glasses" and applied to an image of a "woman without glasses", the resulting image is that of a "woman with glasses". To experiment with this property we generate 1 million sentences from the ARAE-GAN and compute vector transforms in this space to attempt to change main verbs, subjects and modifier (details in Appendix 11). Some examples of successful transformations are shown in Figure 4 (right). Quantitative evaluation of the success of the vector transformations is given in Figure 4 (left).

## 7 CONCLUSION

We present adversarially regularized autoencoders, as a simple approach for training a discrete structure autoencoder jointly with a code-space generative adversarial network. The model learns a improved autoencoder as demonstrated by semi-supervised experiments and improvements on text

transfer experiments. It also learns a useful generative model for text that exhibits a robust latent space, as demonstrated by natural interpolations and vector arithmetic. We do note that (as has been frequently observed when training GANs) our model seemed to be quite sensitive to hyperparameters. Finally, while many useful models for text generation already exist, text GANs provide a qualitatively different approach influenced by the underlying latent variable structure. We envision that such a framework could be extended to a conditional setting, combined with other existing decoding schemes, or used to provide a more interpretable model of language.

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

## 8 APPENDIX: OPTIMALITY PROPERTY

One can interpret the ARAE framework as a dual pathway network mapping two distinct distributions into a similar one; $\text{enc}_\phi$ and $g_\theta$ both output code vectors that are kept similar in terms of Wasserstein distance as measured by the critic. We provide the following proposition showing that under our parameterization of the encoder and the generator, as the Wasserstein distance converges, the encoder distribution ($\mathbf{c} \sim \mathbb{P}_r$) converges to the generator distribution ($\tilde{\mathbf{c}} \sim \mathbb{P}_g$), and further, their moments converge.

This is ideal since under our setting the generated distribution is simpler than the encoded distribution, because the input to the generator is from a simple distribution (e.g. spherical Gaussian) and the generator possesses less capacity than the encoder. However, it is not so simple that it is overly restrictive (e.g. as in VAEs). Empirically we observe that the first and second moments do indeed converge as training progresses (Section 6.1).

**Proposition 1.** *Let $\mathbb{P}$ be a distribution on a compact set $\chi$, and $(\mathbb{P}_n)_{n \in N}$ be a sequence of distributions on $\chi$. Further suppose that $W(\mathbb{P}_n, \mathbb{P}) \to 0$. Then the following statements hold:*

*(i)* $\mathbb{P}_n \rightsquigarrow \mathbb{P}$ *(i.e. convergence in distribution).*

*(ii)* *All moments converge, i.e. for all $k > 1, k \in \mathbb{N}$,*

$$\mathbb{E}_{X \sim \mathbb{P}_n}\Big[\prod_{i=1}^{d} X_i^{p_i}\Big] \to \mathbb{E}_{X \sim \mathbb{P}}\Big[\prod_{i=1}^{d} X_i^{p_i}\Big]$$

*for all $p_1, \ldots, p_d$ such that $\sum_{i=1}^{d} p_i = k$*

*Proof.* (i) has been proved in Villani (2008) Theorem 6.9.

For (ii), using *The Portmanteau Theorem*, (i) is equivalent to:

$\mathbb{E}_{X \sim \mathbb{P}_n}[f(X)] \to \mathbb{E}_{X \sim \mathbb{P}}[f(X)]$ for all bounded and continuous function $f \colon \mathbb{R}^d \to \mathbb{R}$, where $d$ is the dimension of the random variable.

The $k$-th moment of a distribution is given by

$$\mathbb{E}\Big[\prod_{i=1}^{d} X_i^{p_i}\Big] \text{ such that } \sum_{i=1}^{d} p_i = k$$

Our encoded code is bounded as we normalize the encoder output to lie on the unit sphere, and our generated code is also bounded to lie in $(-1, 1)^n$ by the $\tanh$ function. Hence $f(X) = \prod_{i=1}^{d} X_i^{q_i}$ is a bounded continuous function for all $q_i > 0$. Therefore,

$$\mathbb{E}_{X \sim \mathbb{P}_n}\Big[\prod_{i=1}^{d} X_i^{p_i}\Big] \to \mathbb{E}_{X \sim \mathbb{P}}\Big[\prod_{i=1}^{d} X_i^{p_i}\Big]$$

where $\sum_{i=1}^{d} p_i = k$ $\qquad\qquad\square$

## 9 APPENDIX: SHEET OF STYLE-TRANSFER SAMPLES

YELP TRANSFER

| | Positive to Negative | | Negative to Positive |
|---|---|---|---|
| Original | great indoor mall . | Original | hell no ! |
| ARAE | no smoking mall . | ARAE | hell great ! |
| Cross-AE | terrible outdoor urine . | Cross-AE | incredible pork ! |
| Original | great blooming onion . | Original | highly disappointed ! |
| ARAE | no receipt onion . | ARAE | highly recommended ! |
| Cross-AE | terrible of pie . | Cross-AE | highly clean ! |
| Original | i really enjoyed getting my nails done by peter . | Original | bad products . |
| ARAE | i really needed getting my nails done by now . | ARAE | good products . |
| Cross-AE | i really really told my nails done with these things . | Cross-AE | good prices . |
| Original | definitely a great choice for sushi in las vegas ! | Original | i was so very disappointed today at lunch . |
| ARAE | definitely a _num_ star rating for _num_ sushi in las vegas . | ARAE | i highly recommend this place today . |
| Cross-AE | not a great choice for breakfast in las vegas vegas ! | Cross-AE | i was so very pleased to this . |
| Original | the best piece of meat i have ever had ! | Original | i have n't received any response to anything . |
| ARAE | the worst piece of meat i have ever been to ! | ARAE | i have n't received any problems to please . |
| Cross-AE | the worst part of that i have ever had had ! | Cross-AE | i have always the desert vet . |
| Original | really good food , super casual and really friendly . | Original | all the fixes were minor and the bill ? |
| ARAE | really bad food , really generally really low and decent food . | ARAE | all the barbers were entertaining and the bill did n't disappoint . |
| Cross-AE | really good food , super horrible and not the price . | Cross-AE | all the flavors were especially and one ! |
| Original | it has a great atmosphere , with wonderful service . | Original | small , smokey , dark and rude management . |
| ARAE | it has no taste , with a complete jerk . | ARAE | small , intimate , and cozy friendly staff . |
| Cross-AE | it has a great horrible food and run out service . | Cross-AE | great , , , chips and wine . |
| Original | their menu is extensive , even have italian food . | Original | the restaurant did n't meet our standard though . |
| ARAE | their menu is limited , even if i have an option . | ARAE | the restaurant did n't disappoint our expectations though . |
| Cross-AE | their menu is decent , i have gotten italian food . | Cross-AE | the restaurant is always happy and knowledge . |
| Original | everyone who works there is incredibly friendly as well . | Original | you could not see the stage at all ! |
| ARAE | everyone who works there is incredibly rude as well . | ARAE | you could see the difference at the counter ! |
| Cross-AE | everyone who works there is extremely clean and as well . | Cross-AE | you could definitely get the fuss ! |
| Original | there are a couple decent places to drink and eat in here as well . | Original | room is void of all personality , no pictures or any sort of decorations . |
| ARAE | there are a couple slices of options and _num_ wings in the place . | ARAE | room is eclectic , lots of flavor and all of the best . |
| Cross-AE | there are a few night places to eat the car here are a crowd . | Cross-AE | it 's a nice that amazing , that one 's some of flavor . |
| Original | if you 're in the mood to be adventurous , this is your place ! | Original | waited in line to see how long a wait would be for three people . |
| ARAE | if you 're in the mood to be disappointed , this is not the place . | ARAE | waited in line for a long wait and totally worth it . |
| Cross-AE | if you 're in the drive to the work , this is my place ! | Cross-AE | another great job to see and a lot going to be from dinner . |
| Original | we came on the recommendation of a bell boy and the food was amazing . | Original | the people who ordered off the menu did n't seem to do much better . |
| Cross-AE | we came on the recommendation and the food was a joke . | ARAE | the people who work there are super friendly and the menu is good . |
| Cross-AE | we went on the car of the time and the chicken was awful . | Cross-AE | the place , one of the office is always worth you do a business . |
| Original | service is good but not quick , just enjoy the wine and your company . | Original | they told us in the beginning to make sure they do n't eat anything . |
| ARAE | service is good but not quick , but the service is horrible . | ARAE | they told us in the mood to make sure they do great food . |
| Cross-AE | service is good , and horrible , is the same and worst time ever . | Cross-AE | they 're us in the next for us as you do n't eat . |
| Original | the steak was really juicy with my side of salsa to balance the flavor . | Original | the person who was teaching me how to control my horse was pretty rude . |
| ARAE | the steak was really bland with the sauce and mashed potatoes . | ARAE | the person who was able to give me a pretty good price . |
| Cross-AE | the fish was so much , the most of sauce had got the flavor . | Cross-AE | the owner 's was gorgeous when i had a table and was friendly . |
| Original | other than that one hell hole of a star bucks they 're all great ! | Original | he was cleaning the table next to us with gloves on and a rag . |
| ARAE | other than that one star rating the toilet they 're not allowed . | ARAE | he was prompt and patient with us and the staff is awesome . |
| Cross-AE | a wonder our one came in a _num_ months , you 're so better ! | Cross-AE | he was like the only thing to get some with with my hair . |

Table 6: Full sheet of sentiment transfer result

YAHOO TRANSFER

| | from Science | | from Music | | from Politics |
|---|---|---|---|---|---|
| Original | what is an event horizon with regards to black holes ? | Original | do you know a website that you can find people who want to join bands ? | Original | republicans : would you vote for a cheney / satan ticket in 2008 ? |
| Music | what is your favorite sitcom with adam sandler ? | Science | do you know a website that can help me with science ? | Science | guys : how would you solve this question ? |
| Politics | what is an event with black people ? | Politics | do you think that you can find a person who is in prison ? | Music | guys : would you rather be a good movie ? |
| Original | what did john paul jones do in the american revolution ? | Original | do people who quote entire poems or song lyrics ever actually get chosen best answer ? | Original | if i move to the usa do i lose my pension in canada ? |
| Music | what did john lennon do in the new york family ? | Science | do you think that scientists learn about human anatomy and physiology of life ? | Science | if i move the <unk> in the air i have to do my math homework ? |
| Politics | what did john mccain do in the next election ? | Politics | do people who knows anything about the recent issue of <unk> leadership ? | Music | if i move to the music do you think i feel better ? |
| Original | can anybody suggest a good topic for a statistical survey ? | Original | from big brother , what is the girls name who had <unk> in her apt ? | Original | what is your reflection on what will be our organizations in the future ? |
| Music | can anybody suggest a good site for a techno ? | Science | in big bang what is the <unk> of <unk> , what is the difference between <unk> and <unk> ? | Science | what is your opinion on what will be the future in our future ? |
| Politics | can anybody suggest a good topic for a student visa ? | Politics | is big brother in the <unk> what do you think of her ? | Music | what is your favorite music videos on the may i find ? |
| Original | can a kidney infection effect a woman 's <unk> cycle ? | Original | where is the tickets for the filming of the suite life of zack and cody ? | Original | wouldn 't it be fun if we the people veto or passed bills ? |
| Music | can anyone give me a good film <unk> ? | Science | where is the best place of the blood stream for the production of the cell ? | Science | isnt it possible to be cloned if we put the moon or it ? |
| Politics | can a landlord officer have a <unk> <unk> ? | Politics | where is the best place of the navy and the senate of the union ? | Music | isnt it possible or if we 're getting married ? |
| Original | where does the term " sweating <unk> " come from ? | Original | the <unk> singers was a band in 1963 who had a hit called <unk> man ? | Original | can anyone tell me how i could go about interviewing north vietnamese soldiers ? |
| Music | where does the term " <unk> " come from ? | Science | the <unk> river in a <unk> was created by a <unk> who was born in the last century ? | Science | can anyone tell me how i could find how to build a robot ? |
| Politics | where does the term " <unk> " come from ? | Politics | the <unk> are <unk> in a <unk> who was shot an <unk> ? | Music | can anyone tell me how i could find out about my parents ? |
| Original | what other <unk> sources are there than burning fossil fuels . | Original | what is the first metal band in the early 60 's ..... ? ? ? ? | Original | if the us did not exist would the world be a better place ? |
| Music | what other <unk> are / who are the greatest guitarist currently on tv today ? | Science | what is the first country in the universe ? | Science | if the world did not exist , would it be possible ? |
| Politics | what other <unk> are there for veterans who lives ? | Politics | who is the first president in the usa ? ? ? ? ? ? ? ? ? ? ? ? ? ? ? ? ? ? ? ? ? ? ? | Music | if you could not have a thing who would it be ? |

Table 7: Full sheet of Yahoo titles transfer result

| | from Science | | from Music | | from Politics |
|---|---|---|---|---|---|
| Original | take 1ml of hcl ( concentrated ) and dilute it to 50ml . | Original | all three are fabulous artists , with just incredible talent ! ! | Original | 4 years of an idiot in office + electing the idiot again = ? |
| Music | take em to you and shout it to me | Science | all three are genetically bonded with water , but just as many substances , are capable of producing a special case . | Science | 4 years of an idiot in the office of science ? |
| Politics | take bribes to islam and it will be punished . | Politics | all three are competing with the government , just as far as i can . | Music | 4 ) <unk> in an idiot , the idiot is the best of the two points ever ! |
| Original | oils do not do this , they do not " set " . | Original | she , too , wondered about the underwear outside the clothes . | Original | send me $ 100 and i 'll send you a copy - honest . |
| Music | cucumbers do not do this , they do not " do " . | Science | she , too , i know , the clothes outside the clothes . | Science | send me an email and i 'll send you a copy . |
| Politics | corporations do not do this , but they do not . | Politics | she , too , i think that the cops are the only thing about the outside of the u.s. . . | Music | send me $ 100 and i 'll send you a copy . |
| Original | the average high temps in jan and feb are about 48 deg . | Original | i like rammstein and i don 't speak or understand german . | Original | wills can be <unk> , or typed and signed without needing an attorney . |
| Music | the average high school in seattle and is about 15 minutes . | Science | i like googling and i don 't understand or speak . | Science | euler can be <unk> , and without any type of operations , or <unk> . |
| Politics | the average high infantry division is in afghanistan and alaska . | Politics | i like mccain and i don 't care about it . | Music | madonna can be <unk> , and signed without opening or <unk> . |
| Original | the light from you lamps would move away from you at light speed | Original | mark is great , but the guest hosts were cool too ! | Original | hungary : 20 january 1945 , ( formerly a member of the axis ) |
| Music | the light from you tube would move away from you | Science | mark is great , but the water will be too busy for the same reason . | Science | nh3 : 20 january , 78 ( a ) |
| Politics | the light from you could go away from your state | Politics | mark twain , but the great lakes , the united states of america is too busy . | Music | 1966 - 20 january 1961 ( a ) 1983 song |
| Original | van <unk> , on the other hand , had some serious issues ... | Original | they all offer terrific information about the cast and characters , ... | Original | bulgaria : 8 september 1944 , ( formerly a member of the axis ) |
| Music | van <unk> on the other hand , had some serious issues . | Science | they all offer insight about the characteristics of the earth , and are composed of many stars . | Science | moreover , 8 $\hat{3}$ + ( x + 7 ) ( x $\hat{2}$ ) = ( a $\hat{2}$ ) |
| Politics | van <unk> , on the other hand , had some serious issues . | Politics | they all offer legitimate information about the invasion of iraq and the u.s. , and all aspects of history . | Music | harrison : 8 september 1961 ( a ) ( 1995 ) |
| Original | just multiply the numerator of one fraction by that of the other . | Original | but there are so many more i can 't think of ! | Original | anyone who doesnt have a billion dollars for all the publicity cant win . |
| Music | just multiply the fraction of the other one that 's just like it . | Science | but there are so many more of the number of questions . | Science | anyone who doesnt have a decent chance is the same for all the other . |
| Politics | just multiply the same fraction of other countries . | Politics | but there are so many more of the can i think of today . | Music | anyone who doesnt have a lot of the show for the publicity . |
| Original | civil engineering is still an umbrella field comprised of many related specialties . | Original | i love zach he is sooo sweet in his own way ! | Original | the theory is that cats don 't take to being tied up but thats <unk> . |
| Music | civil rights is still an art union . | Science | the answer is he 's definitely in his own way ! | Science | the theory is that cats don 't grow up to <unk> . |
| Politics | civil law is still an issue . | Politics | i love letting he is sooo smart in his own way ! | Music | the theory is that dumb but don 't play <unk> to <unk> . |
| Original | h2o2 ( hydrogen peroxide ) naturally decomposes to form o2 and water . | Original | remember the industry is very shady so keep your eyes open ! | Original | the fear they are trying to instill in the common man is based on what ? |
| Music | jackie and brad pitt both great albums and they are my fav . | Science | remember the amount of water is so very important . | Science | the fear they are trying to find the common ancestor in the world . |
| Politics | kennedy and blair hate america to invade them . | Politics | remember the amount of time the politicians are open your mind . | Music | the fear they are trying to find out what is wrong in the song . |
| Original | the quieter it gets , the more white noise you can here . | Original | but can you fake it , for just one more show ? | Original | think about how much planning and people would have to be involved in what happened . |
| Music | the fray it gets , the more you can hear . | Science | but can you fake it , just for more than one ? | Science | think about how much time would you have to do . |
| Politics | the gop gets it , the more you can here . | Politics | but can you fake it for more than one ? | Music | think about how much money and what would be <unk> about in the world ? |
| Original | h2co3 ( carbonic acid ) naturally decomposes to form water and co2 . | Original | i am going to introduce you to the internet movie database . | Original | this restricts the availability of cash to them and other countries too start banning them . |
| Music | phoebe and jack , he 's gorgeous and she loves to get him ! | Science | i am going to investigate the internet to google . | Science | this reduces the intake of the other molecules to produce them and thus are too large . |
| Politics | nixon ( captured ) he lied and voted for bush to cause his country . | Politics | i am going to skip the internet to get you checked . | Music | this is the cheapest package of them too . |

Table 8: Full sheet of Yahoo answers transfer result

## 10 Appendix: Sample Generations

| ARAE-GAN Samples | AE Samples | LM Samples |
|---|---|---|
| A woman preparing three fish . | Two Three woman in a cart tearing over of a tree . | a man walking outside on a dirt road , sitting on the dock . |
| A woman is seeing a man in the river . | A man is hugging and art . | |
| There passes a woman near birds in the air . | The fancy skier is starting under the drag cup in . | A large group of people is taking a photo for Christmas and at night . |
| Some ten people is sitting through their office . | A dog are <unk> a | |
| The man got stolen with young dinner bag . | A man is not standing . | Someone is avoiding a soccer game . |
| Monks are running in court . | The Boys in their swimming . | The man and woman are dressed for a movie . |
| The Two boys in glasses are all girl . | A surfer and a couple waiting for a show . | Person in an empty stadium pointing at a mountain . |
| The man is small sitting in two men that tell a children . | A couple is a kids at a barbecue . | Two children and a little boy are <unk> a man in a blue shirt . |
| The two children are eating the balloon animal . | The motorcycles is in the ocean loading | |
| A woman is trying on a microscope . | I 's bike is on empty | A boy rides a bicycle . |
| The dogs are sleeping in bed . | The actor was walking in a a small dog area . | A girl is running another in the forest . |
| | no dog is young their mother | the man is an indian women . |

Figure 5: Text samples generated from ARAE-GAN, a simple AE, and from a baseline LM trained on the same data. To generate from an AE we fit a multivariate Gaussian to the learned code space and generate code vectors from this Gaussian.

## 11 Appendix: Vector Arithmetic

We generate 1 million sentences from the ARAE-GAN and parse the sentences to obtain the main verb, subject, and modifier. Then for a given sentence, to change the main verb we subtract the mean latent vector ($\mathbf{t}$) for all other sentences with the same main verb (in the first example in Figure 4 this would correspond to all sentences that had "sleeping" as the main verb) and add the mean latent vector for all sentences that have the desired transformation (with the running example this would be all sentences whose main verb was "walking"). We do the same to transform the subject and the modifier. We decode back into sentence space with the transformed latent vector via sampling from $p_\psi(g(\mathbf{z} + \mathbf{t}))$. Some examples of successful transformations are shown in Figure 4 (right). Quantitative evaluation of the success of the vector transformations is given in Figure 4 (left). For each original vector $\mathbf{z}$ we sample 100 sentences from $p_\psi(g(\mathbf{z} + \mathbf{t}))$ over the transformed new latent vector and consider it a match if *any* of the sentences demonstrate the desired transformation. Match % is proportion of original vectors that yield a match post transformation. As we ideally want the generated samples to only differ in the specified transformation, we also calculate the average word precision against the original sentence (Prec) for any match.

## 12 APPENDIX: EXPERIMENTAL DETAILS

MNIST EXPERIMENTS

- The encoder is a three-layer MLP, `784-800-400-100`.
- Additive Gaussian noise is added into **c** which is then fed into the decoder. The standard deviation of that noise is initialized to be $0.4$, and then exponentially decayed to $0$.
- The decoder is a four-layer MLP, `100-400-800-1000-784`
- The autoencoder is optimized by Adam, with learning rate `5e-04`.
- An MLP generator `32-64-100-150-100`, using batch normalization, and ReLU non-linearity.
- An MLP critic `100-100-60-20-1` with weight clipping $\epsilon = 0.05$. The critic is trained by 10 iterations within each GAN loop.
- Both components of GAN is optimized by Adam, with learning rate `5e-04` on the generator, and `5e-05` on the critic.
- Weighing factor $\lambda^{(1)} = 0.2$.

TEXT EXPERIMENTS

- The encoder is an one-layer LSTM with 300 hidden units.
- Gaussian noise into **c** before feeding it into the decoder. The standard deviation of that noise is initialized to be $0.2$, and then exponentially decayed every 100 iterations by a factor of $0.995$.
- The decoder is a one-layer LSTM with 300 hidden units.
- The decoding process at each time step takes the top layer LSTM hidden state and concatenates it with the hidden codes **c**, before feeding them into the output (i.e. vocabulary projection) and the softmax layer.
- The word embedding is of size 300.
- We adopt a grad clipping on the encoder/decoder, with max `grad_norm` = 1.
- The encoder/decoder is optimized by vanilla SGD with learning rate 1.
- An MLP generator `100-300-300`, using batch normalization, and ReLU non-linearity.
- An MLP critic `300-300-1` with weight clipping $\epsilon = 0.01$. The critic is trained by 5 iterations within each GAN loop.
- Both components of GAN are optimized by Adam, with learning rate `5e-05` on the generator, and `1e-05` on the critic.
- We increment the number of GAN training loop[3] by 1 (it initially is set to 1) , respectively at the beginning of epoch #2, epoch #4 and epoch #6.

SEMI-SUPERVISED EXPERIMENTS

Similar to the SNLI generation experiment setup, with the following changes:

- We employ larger network to GAN components: MLP generator `100-150-300-500` and MLP critic `500-500-150-80-20-1` with weight clipping factor $\epsilon = 0.02$. The critic is trained by 10 iterations within each GAN loop.

YELP/YAHOO TRANSFER

Similar to the SNLI setup, with the following changes

- The encoder and decoder size are both increased to 500 hidden units.
- The style adversarial classifier is an MLP with structure `300-200-100`, with learning rate 0.1 trained with SGD.
- We employ both larger generator and discriminator architectures in GAN: generator `200-400-800` with $z$ dim being set to 64; discriminator `300-160-80-20`.
- Weighing factor for critic gradient $\lambda_a^{(1)} = 1$, $\lambda_b^{(1)} = 10$.
- No GAN loop scheduling is employed here.

---

[3]The GAN training loop refers to how many times we train GAN in each entire training loop (one training loop contains training autoencoder for one loop, and training GAN for one or several).

