# OpenReview forum: "Adversarially Regularized Autoencoders"
_ICLR.cc/2018/Conference — Invite to Workshop Track_

### Official Review · AnonReviewer3 · 2017-11-28
**interesting idea; maybe helpful to present more intuition**

**Rating:** 6
**Confidence:** 3

**Review:**

the paper presents a way to encode discrete distributions which is a challenging problem. they propose to use a latent variable gan with one continuous encoding and one discrete encoding.

two questions linger around re practices:
1. gan is known to struggle with discriminating distributions with different supports. the problem also persists here as the gan is discriminating between a continuous and a discrete distribution.  it'll interesting to see how the proposed approach gets around this issue.

2. the second question is related. it is unclear how the optimal distribution would look like with the latent variable gan. ideally, the discrete encoding be simply a discrete approximation of the continuous encoding. but optimization with two latent distributions and one discriminator can be hard. what we get in practice is pretty unclear. also how this could outperform classical discrete autoencoders is unclear. gan is an interesting idea to apply to solve many problems; it'll be helpful to get the intuition of which properties of gan solves the problem in this particular application to discrete autoencoders.

---

> ### Author Response · Authors · 2017-12-05
> **Respond to AnonReviewer3**
>
> Thanks for the review.
>
> We feel there is perhaps a misunderstanding in the review, and apologize if it came from our end. This work uses continuous encodings only. Specifically, we use an encoder to convert discrete sequence (e.g. text) into continuous code space. Our GAN distribution is given by transforming a continuous random variable into the continuous code space. The adversarial training happens only within the continuous space.  If you could revisit the model diagram Figure 1 in the paper, the P_r and P_g are both pointed to the continuous code space. Therefore, the critic is *not* built to discriminate between a continuous and discrete distribution. As such, ARAE does not aim for a discrete approximation of the continuous encoding.
>
> The question regarding the optimal distribution is very interesting. The intuition of ARAE is that it learns a contraction of the discrete sample space into a continuous one through the encoder, and smoothly assigns similar codes c and c' to similar x and x'. We provide some intuition in the text of section 4, and corresponding experiments in section 6.1.
>
> We also want to point out another two submissions to this ICLR that are similar to our paper:  [1] and [2].
>
> [1] Wasserstein Auto-Encoders
> [2] Learning Priors for Adversarial Autoencoders

---

### Official Review · AnonReviewer1 · 2017-11-30
**Good paper but disregards formatting suggestions, making fair evaluation impossible**

**Rating:** 3
**Confidence:** 4

**Review:**

This paper introduces a model for learning robust discrete-space representations with autoencoders. The proposed method jointly trains an RNN encoder with a GAN to produce latent representations which are designed to better encode similarity in the discrete input space. A variety of experiments are conducted that demonstrate the efficacy of the proposed methodology.

Generally speaking, I like the overall idea, which, as far as I know, is a novel approach for dealing with discrete inputs. The generated textual samples look good and offer strong support for the model. However, I would have preferred to see more quantitative evaluation and less qualitative evaluation, but I understand that doing so is challenging in this domain.

I will refrain from adding additional detailed commentary in this review because I am unable to judge this paper fairly with respect to other submissions owing to its large deviation from the suggested length limits. The call for papers states that "we strongly recommend keeping the paper at 8 pages", yet the current submission extends well into its 10th page. In addition (and more importantly), the margins appear to have been reduced relative to the standard latex template. Altogether, it seems like this paper contains a significant amount of additional text beyond what other submissions enjoyed. I see no strong reason why this particular paper needed the extra space. In fact, there are obvious places where the exposition is excessively verbose, and there are clear opportunities to reduce the length of the submission. While I fully understand that the length suggestions are not requirements, in my opinion this paper did not make an adequate effort to abide by these suggestions. Moreover, as a result, I believe this extra length has earned this paper an unfair advantage relative to other submissions, which themselves may have removed important content in order to abide by the length suggestions. As such, I find it difficult or impossible to judge this paper fairly relative to other submissions. I regrettably cannot recommend this paper for acceptance owing to these concerns.

There are many good ideas and experiments in this paper and I would strongly encourage the authors to resubmit this work to a future conference, making sure to reorganize the paper to adhere to the relevant formatting guidelines.

---

> ### Author Response · Authors · 2017-12-05
> **Respond to AnonReviewer1**
>
> Thank you for the review.
>
> While the reviewer accurately notes that the work exceeds 8 pages (it is 9 pages), we believe that unlike other conferences, ICLR purposefully makes this a “suggestion” and not a hard requirement. This interpretation seems to be the clear consensus of the ICLR community. Specifically, in the top-20 reviewed papers listed here: https://chillee.github.io/OpenReviewExplorer/, what we found was: 11 out of 20 papers go over 8 pages; 9 out of 20 papers go over or stay at 9 pages; 3 papers go over 10 pages. Additionally the reviewer claims that we changed the underlying template, which we do not believe is true. Figure 3 and Table 4 extend to the margins, but this seems common as well in other papers.
>
> Given these points we ask for the reviewer to please do a content based assessment of the paper. If they really find that length is an issue, we are happy to move some content to the appendix, but given the above statistics and purposeful relaxness of ICLR rules, it seems arbitrary to reject a paper strictly on formatting terms.

---

### Official Review · AnonReviewer2 · 2017-12-01
**Very nice paper, very clearly presented, on using a learned distribution to regularize the embedding space of a discrete-space autoencoder.**

**Rating:** 9
**Confidence:** 3

**Review:**

The authors present a new variation of autoencoder, in which they jointly train (1) a discrete-space autoencoder to minimize reconstuction loss, and (2) a simpler continuous-space generator function to learn a distribution for the codes, and (3) a GAN formulation to constrain the distributions in the latent space to be similar.

The paper is very clearly written, very clearly presented, addresses an important issue, and the results are solid.

My primary suggestion is that I would like to know a lot more (even qualitatively, does not need to be extensively documented runs) about how sensitive the results were--- and in what ways were they sensitive--- to various hyperparameters. Currently, the authors mention in the conclusion that, as is known to often be the case with GANS, that the results were indeed sensitive. More info on this throughout the paper would be a valuable contribution. Clearly the authors were able to make it work, with good results. When does it not work? Any observations about how it breaks down?

It is interesting how strong the denoising effect is, as simply a byproduct of the adversarial regularization.

Some of the results are quite entertaining indeed. I found the yelp transfer results particularly impressive.

(The transfer from positive->negative on an ambiguous example was interesting: Original "service is good but not quick" -> "service is good but not quick, but the service is horrible", and "service is good, and horrible, is the same and worst time ever". I found it interesting to see what it does with the mixed signals of the word "but": on one hand, keeping it helps preserve the structure of the sentence, but on the other hand, keeping it makes it hard to flip the valence. I guess the most accurate opposite would have been "The service is quick but not good"... )

I really like the reverse perplexity measure. Also, it was interesting how that was found to be high on AAE due to mode-collapse.

Beyond that, I only have a list of very insignificant typos:
-p3, end of S3, "this term correspond to minimizing"
-p3, S4, "to approximate Wasserstein-1 term" --> "to approximate the Wasserstein-1 term"
-Figure 1, caption "which is similarly decoded to $\mathbf{\~x}$" . I would say that it is "similarly decoded to $\mathbf{c}$", since it is \mathbf{c} that gets decoded. Unless the authors meant that it "is similarly decoded to produce $\mathbf{\~x}$. Alternately, I would just say something like "to produce a code vector, which lies in the same space as \mathbf{c}", since the decoding of the generated code vector does not seem to be particularly relevant right here.

-p5, beginning of Section 6.1:  "to regularize the model produce" --> "to regularize the model to produce" ?
-p6, end of first par. "is quite high for the ARAE than in the case" --> quite a bit higher than? etc...
-p7, near the bottom "shown in figure 6". --> table, not figure...
-p8  "ability mimic" -->"ability to mimic"
-p9 Fig 3 -- the caption is mismatched with the figure.. top/bottom/left/right/etc.... Something is confusing there...
-p9 near the bottom "The model learns a improved" --> "The model learns an improved"
-p14 left side, 4th cell up, "Cross-AE"-->"ARAE"

This is a very nice paper with a clear idea (regularize discrete autoencoder using a flexible rather than a fixed prior), that makes good sense and is very clearly presented.

In the words of one of the paper's own examples: "It has a great atmosphere, with wonderful service." :)
Still, I wouldn't mind knowing a little more about what happened in the kitchen...

---

> ### Author Response · Authors · 2017-12-05
> **Respond to AnonReviewer2**
>
> Thanks for the comments.
>
> As has been observed for many GANs, training ARAE required hyperparameter tuning for learning rate, weight clipping factor and the architecture. We found that suboptimal hyperparameter led to mode collapse. We used reverse PPL as a proxy to test for this issue. In this work we used the original setting of WGAN with weight clipping. It is possible that using the updated version WGAN-GP with gradient penalty could help with stability. We will make these points more clear in our next version.
>
> Thank you for the helpful points on the experiments and  for pointing out the typos. We will correct them in the next version.

---

> ### Comment · AnonReviewer2 · 2018-01-19
> **discussion/revision in response to AnonReviewer1 and AnonReviewer4**
>
> For the record, I completely disagree with AnonReviewer1 and completely do agree with the authors' response: the page limit is soft and this submission did not exceed it in any significant way.
>
> I found AnonReviewer4 raised some interesting points and questions.
> I believe that they are generally addressable, to different extents.
> Exposition issues aside, there are two key issues that would give me inclination to change my score, if at all:
> 1) the question of similarity to AAE is perhaps the most important one in terms of revising my score. I do believe the authors' response where they say that they tried it on the same task and it didn't work. I would suggest mentioning this in the paper itself.  e.g. Is it something about the language modelling task where allowing a learnable prior becomes a significant advantage? Is there more to be said about this?
> 2) the other criticism that I find particularly interesting is requesting justification of the reverse-PPL. I still find this metric very interesting in this context and I don’t *require* that justification but I think including it will only strengthen the paper (and the comparison with Parzen windows etc). Given the general lousiness of evaluation methods for generative models, this is an interesting discussion. And again, as with point #1 above, there are differences between what "works" for generating images and generating language, and identifying those differences is worthwhile.
>
> I am still OK with my score of (Score 9, Conf 3), although (Score 8, Conf 3) would work too. My latent score (i.e. in a continuous space) might be around (8.5, 3.5).

---

> > ### Author Response · Authors · 2018-01-20
> > **Reply to "discussion/revision in response to AnonReviewer1 and AnonReviewer4"**
> >
> > Indeed, the discussion around parametrized prior versus the classical prior is very interesting. In this work we only explore this in the universe of autoencoders, i.e., ARAE/AAE. The study of a generic form of parametrized prior may require research on numerous other machine learning framework/schemes, and that in our opinion is beyond the scope of this paper.
> > 1. To address the similarity and dissimilarity between ARAE and AAE: a possible reason for why learning a prior works much better empirically could be the same reason why autoregressive flows perform much better for VAEs. Indeed, Chen et al. 2017 observe improvements by transforming the spherical Gaussian to a more complex prior through parameterized neural networks.
> > 2. Our informal justification is that RNNLM is a *very* good model for scoring text (e.g. it is frequently combined with speech recognition/machine translation systems). So unlike the case with Parzen windows where a good Parzen window score does not necessarily imply good generations (Theis et al. 2016), we think that it will be very hard to game the Reverse PPL metric. Of course, a formal justification would be nice (e.g. if (i) the score between p_rnn(x) from an RNNLM and p_star(x) from the true distribution is within some bound epsilon1 for all x and (ii) reverse PPL from real data vs generated data is smaller than another bound epsilon2, then KL(p_theta(x) || p_star(x)) is below some bound delta that is a function of epsilon1 and epsilon2), but perhaps beyond the scope of this paper.
> >
> > Lastly did you mean you were fine with (9, conf 3) and (8, conf 4)? Because the interpolation of that gives (8.5, conf 3.5) in the latent space :)

---

### Official Review · AnonReviewer4 · 2018-01-17
**Little methodological novelty but an interesting set of tasks considered, empirical evaluation could still be better.**

**Rating:** 5
**Confidence:** 4

**Review:**

I was asked to contribute this review rather late in the process, and in order
to remain unbiased I avoided reading other reviews. I apologize if some of
these comments have already been addressed in replies to other reviewers.

This paper proposes a regularization strategy for autoencoders that is very
similar to the adversarial autoencoder of Makhzani et al. The main difference
appears to be that rather than using the classic GAN loss to shape the
aggregate posterior of an autoencoder to match a chosen, fixed distribution,
they instead employ a Wasserstein GAN loss (and associated weight magnitude
constraint, presumably enforced with projected gradient descent) on a system
where the matched distribution is instead learned via a parameterized sampler
("generator" in the GAN lingo). Gradient steps that optimize the encoder,
decoder and generator are interleaved. The authors apply an extension of this
method to topic and sentiment transfer and show moderately good latent space
interpolations between generated sentences.

The difference from the original AAE is rather small and straightforward, making the
novelty mainly in the choice of task, focusing on discrete vectors and sequences.

The exposition leaves ample room for improvement. For one thing, there is the
irksome and repeated use of "discrete structure" when discrete *sequences* are
considered almost exclusively (with the exception of discretized MNIST digits).
The paper is also light on discussion of related work other than Makhzani et al
-- the wealth of literature on combining autoencoders (or autoencoder-like
structures such as ALI/BiGAN) and GANs merits at least passing mention.

The empirical work is somewhat compelling, though I am not an expert in this
task domain. The annealed importance sampling technique of Wu et al (2017) for
estimating bounds on a generator's log likelihood could be easily applied in
this setting and would give (for example, on binarized MNIST) a quantitative
measurement of the degree of overfitting, and this would have been preferable
than inventing new heuristic measures. The "Reverse PPL" metric requires more
justification, and it looks an awful lot like the long-since-discredited Parzen
window density estimation technique used in the original GAN paper.

High-level comments:

- It's not clear why the optimization is done in 3 separate steps. Aside
from the WGAN critic needing to be optimized for more steps, couldn't the
remaining components be trained jointly, with a weighted sum of terms for the
encoder?
- In section 2, "This [pre-training or co-training with maximum likelihood]
  precludes there being a latent encoding of the sentence." It is not at all
  clear to me why this would be the case.
- "One benefit of the ARAE framework is that it compresses the input to a
  single code vector." This is true of any autoencoder.
- It would be worth explaining, in a sentence, the approach in Shen et al for
  those who are not familiar with it, seeing as it is used as a baseline.
- We are told that the encoder's output is l2-normalized but the generator's
  is not, instead output units of the generator are squashed with the tanh
  activation. The motivation for this choice would be helpful. Shortly
  thereafter we are told that the generator quickly learns to produce norm 1
  outputs as evidence that it is matching the encoder's distribution, but this
  is something that could have just as easily have been built-in, and is a
  trivial sort of "distribution matching"
- In general, tables that report averages would do well to report error bars as
  well. In general some more nuanced statistical analysis of these results
  would be worthwhile, especially where they concern human ratings.
- The dataaset fractions chosen for the semi-supervised experience seem
  completely arbitrary. Is this protocol derived from some other source?
  Putting these in a table along with the results would improve readability.
- Linear interpolation in latent space may not be the best choice here
  seeing as e.g. for a Gaussian code the region near the origin has rather low
  probability. Spherical interpolation as recommended by White (2016) may
  improve qualitative results.
- For the interpolation results you say "we output the argmax", what is meant?
  Is beam search performed in the case of sequences?
- Finally, a minor point: I will challenge the authors to justify their claim
  that the learned generative model is "useful" (their word). Interpolating
  between two sentences sampled from the prior is a neat parlour trick, but the
  model as-is has little utility. Even some speculation on how this aspect
  could be applied would be appreciated (admittedly, many GAN papers could use
  some reflection of this sort).

---

> ### Author Response · Authors · 2018-01-18
> **Respond to AnonReviewer4 (part 1)**
>
> We thank the reviewer for a very thoughtful review. Before responding to more specific points, we want to point out that unlike the case with images where established architectures/baselines/metrics exist (e.g. DCGAN), GANs for text is still very much an open problem, and there is no consensus on which approach works best (policy gradients/Gumbel-softmax, etc). Given the current exploratory landscape of text GANs, we believe that our work represents a simple but interesting alternative to other approaches, backed up by quantitative and qualitative experiments. We therefore ask the reviewer to kindly reconsider the work in the context of existing work on GANs for text.
>
> Specific points:
>
> -  The difference from the original AAE is rather small and straightforward, making the
> novelty mainly in the choice of task, focusing on discrete vectors and sequences.
>
> Response: Indeed, from a methodological standpoint our method is similar to the AAE. However, this small difference (i.e. learning a prior through a parameterized generator) was crucial in making the model work. When we tried training AAEs for this dataset and we observed severe mode-collapse (reverse PPL: ~900).
>
> - The exposition leaves ample room for improvement. For one thing, there is the
> irksome and repeated use of "discrete structure" when discrete *sequences* are
> considered almost exclusively (with the exception of discretized MNIST digits).
> The paper is also light on discussion of related work other than Makhzani et al
> -- the wealth of literature on combining autoencoders (or autoencoder-like
> structures such as ALI/BiGAN) and GANs merits at least passing mention.
>
> Response: Thank you for pointing this out. We will change the wording for more clarity. We will also add more discussion regarding ALI/BiGAN. We do want to point out however that while these works are similar in that they work with (x,z) space, they typically perform discrimination/generation in the joint (x,z) space, and therefore would face difficulties when applied directly to discrete spaces.
>
> - The empirical work is somewhat compelling, though I am not an expert in this
> task domain. The annealed importance sampling technique of Wu et al (2017) for
> estimating bounds on a generator's log likelihood could be easily applied in
> this setting and would give (for example, on binarized MNIST) a quantitative
> measurement of the degree of overfitting, and this would have been preferable
> than inventing new heuristic measures. The "Reverse PPL" metric requires more
> justification, and it looks an awful lot like the long-since-discredited Parzen
> window density estimation technique used in the original GAN paper.
>
> Response: Our understanding is that using log-likelihood estimates from Parzen windows is bad because Parzen windows are (very) bad models of images. In contrast, an RNN LM has been well-established to be quite good (in fact, state-of-the-art) at *scoring* text. We thus believe that Reverse PPL is a fair metric for quantitatively assessing generative models of text, despite its ostensible similarity/motivation to Parzen windows. The AIS technique from Wu et al. (2017) would not be applicable in our case because we need to be able to calculate p_\theta(x_test)  (Wu et al. (2017) actually use Parzen windows combined with AIS to give log-likelihood estimates of GAN-based models).
>
> - It's not clear why the optimization is done in 3 separate steps. Aside
> from the WGAN critic needing to be optimized for more steps, couldn't the
> remaining components be trained jointly, with a weighted sum of terms for the
> encoder?
>
> Response: We optimized the objectives separately as this is the standard setup in GAN training. The remaining objectives could indeed be trained jointly, but we did not try this.

---

> > ### Author Response · Authors · 2018-01-18
> > **Respond to AnonReviewer4 (part 2)**
> >
> > - In section 2, "This [pre-training or co-training with maximum likelihood]
> >   precludes there being a latent encoding of the sentence." It is not at all
> >   clear to me why this would be the case.
> >
> > Response: When pre-training/co-training with a language model, there is no latent vector z, as the language model objective is given by log p(x) = \sim_{t=1}^T logp(x_t | x_{<t}) (i.e. p(x_t) just depends on the previous tokens x_{<t}, not a latent vector).
> >
> > - "One benefit of the ARAE framework is that it compresses the input to a
> >   single code vector." This is true of any autoencoder.
> >
> > Response: Right. We wanted to emphasize the fact that having a fixed-dimensional vector representation of a sentence allows for simpler manipulations in the latent space (compared to, for example, sequential VAEs (Chung et al. 2015) that have a latent vector for each time step). We will change the wording to get this point across better.
> >
> > - It would be worth explaining, in a sentence, the approach in Shen et al for
> >   those who are not familiar with it, seeing as it is used as a baseline.
> >
> > Response: Good point! We will describe Shen et al. in more detail.
> >
> > - We are told that the encoder's output is l2-normalized but the generator's
> >   is not, instead output units of the generator are squashed with the tanh
> >   activation. The motivation for this choice would be helpful. Shortly
> >   thereafter we are told that the generator quickly learns to produce norm 1
> >   outputs as evidence that it is matching the encoder's distribution, but this
> >   is something that could have just as easily have been built-in, and is a
> >   trivial sort of "distribution matching"
> >
> > Response: Mainly based on empirical experiments: l2-normalized output from the encoder stabilizes training; squashing the output from the generator by tanh was adopted from DCGAN. We will try to discuss more on this via experiments in the next revision.
> >
> > - In general, tables that report averages would do well to report error bars as
> >   well. In general some more nuanced statistical analysis of these results
> >   would be worthwhile, especially where they concern human ratings.
> >
> > Response: We will add error bars for the various measures where applicable (e.g. Human ratings). Some metrics are inherently at the corpus level and thus error bar estimation is not so straightforward.
> >
> > - The dataset fractions chosen for the semi-supervised experience seem
> >   completely arbitrary. Is this protocol derived from some other source?
> >   Putting these in a table along with the results would improve readability.
> >
> > Response: This was arbitrary. We will make it clearer in the table/text.
> >
> > - Linear interpolation in latent space may not be the best choice here
> >   seeing as e.g. for a Gaussian code the region near the origin has rather low
> >   probability. Spherical interpolation as recommended by White (2016) may
> >   improve qualitative results.
> >
> > Response: Yes, spherical interpolation is an interesting alternative and we can certainly try it out. Given the relatively low dimension z-space however, people have found simple linear interpolation to work well enough in images.
> >
> > - For the interpolation results you say "we output the argmax", what is meant?
> >   Is beam search performed in the case of sequences?
> >
> > Response: We perform greedy decoding. We will make this clearer.
> >
> > - Finally, a minor point: I will challenge the authors to justify their claim
> >   that the learned generative model is "useful" (their word). Interpolating
> >   between two sentences sampled from the prior is a neat parlour trick, but the
> >   model as-is has little utility. Even some speculation on how this aspect
> >   could be applied would be appreciated (admittedly, many GAN papers could use
> >   some reflection of this sort).
> >
> > Response: We completely agree! Usefulness of GANs (whether on images/text) is still an open issue and could definitely use more reflection. But recent results on unaligned transfer (DiscoGAN/CycleGAN/Text Style transfer/Unsupervised NMT), including the results presented in this work, give a compelling case for the utility of latent representations learned via adversarial training. We will make sure to temper the language to reflect the preliminary nature of work in this area though.

---

### Decision · Program_Chairs · 2018-01-29
**ICLR 2018 Conference Acceptance Decision**

**Decision:**

Invite to Workshop Track

**Comment:**

In general, the reviewers and myself find this work of some interest, though potentially somewhat incremental in terms of technical novelty compared to the work for Makhzani et al. Another bothersome aspect is the question of evaluation and understanding how well the model actually does; I am not convinced that the interpolation experiments are actually giving us a lot of insights. One interesting ablation experiment (suggested privately by one of the reviewers) would be to try AAE with Wasserstein and without a learned generator -- this would disambiguate which aspects of the proposed method bring most of the benefit. As it stands, the submission is just shy of the acceptance bar, but due to its interesting results in the natural language domain, I do recommend it being presented at the workshop track.